# On Memorization of Large Language Models in Logical Reasoning

**Chulin Xie**[‡]    **Yangsibo Huang**[†,¶]    **Chiyuan Zhang**[†]
**Da Yu**[†]    **Xinyun Chen**[†]    **Bill Yuchen Lin**[§]    **Bo Li**[‡]    **Badih Ghazi**[†]    **Ravi Kumar**[†]
[†]Google    [‡]University of Illinois Urbana-Champaign    [¶]Princeton University    [§]Allen Institute for AI

## Abstract

Large language models (LLMs) show good performance on some complicated reasoning tasks, yet could also make the most basic reasoning mistakes. This contrasting behavior is puzzling when it comes to understanding the mechanisms behind LLMs' reasoning capabilities. One hypothesis is that the increasingly high and nearly saturated performance on common reasoning benchmarks could be due to the memorization of similar benchmark problems accidentally leaked into the training data. In this paper, we systematically investigate this problem with a measurement of memorization in reasoning tasks inspired by human behaviors, and a dynamically generated logical reasoning benchmark based on Knights and Knaves puzzles. We found that LLMs could interpolate the training puzzles (achieving $\sim 100\%$ accuracy) after fine-tuning, yet fail when those puzzles are slightly perturbed, suggesting that the models heavily rely on memorization to solve those training puzzles. On the other hand, we show that LLMs learn to reason while interpolating the training set. At higher level of memorization, the model not only solves more unseen test puzzles, but also solves them relatively robustly under perturbation. This phenomenon suggests that LLMs exhibit a complex interplay between memorization and genuine reasoning abilities, and reveals an interesting direction for future research. Our code and data are available at https://memkklogic.github.io/.

## 1   Introduction

Modern Large Language Models (LLMs) show impressive reasoning capabilities that allow them to solve a wide range of problems including commonsense reasoning and mathematical reasoning. In the meantime, LLMs also make mistakes on some of the most basic problems (e.g., comparing which number is bigger – 13.11 or 13.8 [17], and counting the number of sisters that Alice's brother have [21]).

Their contrast of both superhuman reasoning capabilities and dumb mistakes is puzzling when it comes to understanding how exactly LLMs perform reasoning tasks. This question is important both scientifically and practically: understanding how LLMs reason could shed light on our understanding of learning and generalization behaviors of LLMs; and is crucial for real-world applications where robust reasoning is required to mitigate safety and trustworthiness concerns [25, 15, 26].

One hypothesis is that LLMs could be relying on *memorization* when solving those reasoning tasks, especially when measured by popular benchmark datasets that could be accidentally leaked into the massive internet-crawled pre-training datasets. Previous work [5, 24] show that LLMs could indeed memorize the training data. However, most of those studies focus on analyzing memorization from the perspective of privacy [6] or copyright [13, 27] concerns. Other papers focus on designing dynamic benchmarks [32, 23, 22, 11] or alternative evaluation protocols [30, 31, 28, 23] that could mitigate the issue of benchmark saturation potentially due to memorization. In this paper, we take

NeurIPS 2024 Workshop on Mathematical Reasoning and AI.

a direct approach to quantify memorization in reasoning tasks and analyze the interplay between memorization and reasoning. Specifically, we summarize our contributions below:

- To quantify memorization in reasoning tasks, we define a memorization metric based on the notions of interpolation and the performance inconsistency under local perturbation that are inspired by human behaviors.

- To facilitate the measurement, we propose a new logical reasoning benchmark based on the *Knights and Knaves* (K&K) [12] puzzles, that support the automatic generation of new puzzles with different difficulty levels and local perturbations of existing puzzles.

- We show that K&K puzzles are challenging and only the most advanced LLMs could consistently solve them. The generally low accuracy observed across most off-the-shelf models indicates that K&K puzzles are likely uncommon in internet-based training data. However, our analysis suggests that certain models exhibit signs of memorization to solve the puzzles.

- By fine-tuning on K&K samples, we confirm that modern LLMs are capable of memorizing a large collection of puzzles and their solutions when seen during training. Interestingly, when measuring accuracy on the unseen test puzzles, we found that the models' reasoning capabilities *grow* with the amount of memorization as the models *interpolate* [3, 20, 2, 1] the training set[1]. Additionally, these enhanced reasoning abilities transfer across different levels of puzzle difficulty.

## 2 How to Measure Memorization in Reasoning Tasks

### 2.1 Memorization Metrics for Reasoning Tasks

Memorization of LLMs has been studied in various contexts such as privacy, copyright [6, 13, 27], and solving knowledge intensive tasks [3, 10]. In this paper, we are specifically interested in measuring the level of memorization when solving reasoning tasks. This kind of behaviors can be observed on humans. For example, when preparing for an exam / interview, people may not be able to fully digest the underlying principles due to various reasons or constraints. But when (luckily) facing the same problem one has prepared for, they would still be able to solve it. The key characteristics of this type of memorization are: (A) high accuracy on observed problems; (B) low accuracy on unseen but similar problems, due to the lack of deep understanding.

Based on this intuition, for a dataset $\mathcal{D}$ of reasoning puzzles, we measure the following two quantities:

1. To measure (A), we use the accuracy $\mathsf{Acc}(f; \mathcal{D})$ to measure the percentage of the puzzles in $\mathcal{D}$ that $f$ can solve. We are especially interested in measuring on the set of *observed puzzles*, i.e. the training set, $\mathsf{Acc}(f; \mathsf{Tr})$. We say $f$ **interpolates** [3, 20, 2, 1] the training puzzles if $\mathsf{Acc}(f; \mathsf{Tr}) \approx 100\%$.

2. To measure (B), we measure a *consistency ratio* $\mathsf{CR}(f; \mathcal{D})$ between the number of *consistently solved puzzles* after some *local perturbations*, and the number of solved puzzles (without perturbation). We are interested in local perturbations that makes minimal change to the puzzle and maintain the difficulty level (to be specified in § 2.2).

We combine the two factors to define a **Local Inconsistency based Memorization Score**:

$$\mathsf{LiMem}(f; \mathcal{D}) = \mathsf{Acc}(f; \mathcal{D}) \cdot (1 - \mathsf{CR}(f; \mathcal{D})).$$

When there is no ambiguity, we simply call it the memorization score. $\mathsf{LiMem}(f; \mathcal{D}) \in [0, 100]\%$ and a larger score provide a stronger evidence of memorization. In our empirical study, we say $f$ solves $\mathcal{D}$ **via memorization** if $\mathsf{LiMem}(f; \mathcal{D}) > 10\%$; otherwise we say $f$ solves $\mathcal{D}$ **via reasoning**. Specifically, a high $\mathsf{LiMem}(f; \mathsf{Tr})$ matches the characteristic behavior of human memorizing observed puzzles, and in this case we say $f$ **memorized the training puzzles**. Furthermore, we also measure $\mathsf{LiMem}(f; \mathsf{Tst})$ on test examples, to study if the generalization accuracy is due to reasoning or memorization.

In order to effectively measure memorization score $\mathsf{LiMem}(f; \mathcal{D})$, we need a principled way to (1) perform local perturbation that changes the problem while maintaining its difficulty level; (2) compute the new answer after perturbation, which should be different from the original answer. Towards this goal, we design and implement a functional dataset based on the Knights and Knaves puzzles [12].

---

[1]Interpolating is a term in learning theory to indicate fitting 100% accuracy on training set.

## 2.2 Knights and Knaves Logical Reasoning Benchmark

Knights and Knaves (K&K) is a type of logic puzzle where some characters can only answer questions truthfully, and others only falsely. The goal is to infer each person's identity. For example: *A very special island is inhabited only by knights and knaves. Knights always tell the truth, and knaves always lie. You meet 2 inhabitants: Samuel, and Isabella. Samuel told you that Isabella is a knight. Isabella said that Samuel is a knave and Isabella is a knight. So who is a knight and who is a knave?* The ground-truth answer is that *(1) Samuel is a knave and (2) Isabella is a knave.*

Based on the K&K puzzle, we design a *dynamic* benchmark that supports generating new problems and perturbing existing problems. Our library automatically solves the K&K puzzles and generates solutions for evaluation and training. Specifically, our benchmark consists of two components:

**The abstract problem sampler** generates random K&K puzzles in an abstract format (see details in § B). Specifically, it takes as input the problem specification $(N, D, W)$ that determines the difficulty level. It then generates a problem with $N$ persons, and for each person, a statement that consists of a random tree of maximum width $W$ and depth $D$. The leaf nodes can be a claim that a specific person is lying (i.e., knaive) or telling the truth (i.e., knight) , whereas the branching node can be *and*, *or*, *not*, *if*, and *if-and-only-if*. The problem sampler also has two subcomponents: the **Solver** finds all possible solutions to a given puzzle, which is used to guarantee that we generate only problems with a unique solution; the **Perturber** which, given a problem, generates a locally perturbed version, by replacing a leaf node or an entire statement of a random person's statement with a different one. Perturber only keeps the perturbed problems that have a different solution than the original problem. Comparison between the original sample and the leaf/statement-perturbed samples is provided in Tab. 1.

**The natural language generator** takes an abstract K&K problem and formats it in natural language. The formatting is template-based, but we support a variety of different (common and uncommon) person names, role names (e.g., knight & knaves, angels & devils), and different styles of making each person's claim.

We create disjoint sets of $n_{\text{train}}$ training problems and $n_{\text{test}}$ testing problems for each $N$-person task. Here, $n_{\text{test}} = 100$, $n_{\text{train}} = 1,000$ for $N > 2$, and $n_{\text{train}} = 200$ for 2-person tasks due to limited combinations. Then, we generate perturbed versions for each problem.

# 3 Quantifying LLM Memorization of Reasoning Tasks

## 3.1 Off-the-shelf Models

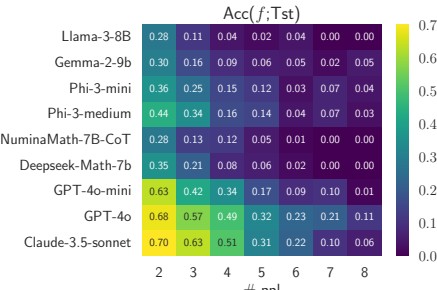

Figure 1: Under 0-shot direct prompting, test accuracy of off-the-shelf models drops significantly with increasing puzzle complexity.

We use the K&K benchmark (§ 2.2) to evaluate 8 models that are shown to perform competitively on common reasoning benchmarks. We utilize zero-shot direct prompting with task-specific instructions for open-ended question-answering. To assess the correctness, we implement keyword matching: a response is considered correct if each person's ground truth identity appears in the conclusion part of the model's output (see more details in § C). As shown in Fig. 1, our K&K benchmark poses a challenging logical reasoning task for all the models. Even for the easiest problems involving only 2 persons, the best models still achieve $< 70\%$ accuracy. And the performance drops significantly as the complexity increases (the best accuracy is only $11\%$ for 8-person problems).

To quantify LLMs' memorization of the logical reasoning task, we employ the metrics proposed in the previous section. Since the training data for the off-the-shelf models is unknown, we will delay the measurement of the interpolation to fine-tuned models in § 3.2 and focus on the memorization score under local perturbation $\text{LiMem}(f; \text{Tst})$ here. As shown in Fig. 1, the test accuracy is relatively low for most cases, suggesting K&K-related problems are probably rare in the Internet and in the training sets of these models. However, we also note that some specific models have large gaps under local perturbation as shown in Fig. 5, such as GPT4o and Claude-3.5-Sonnet on 3-person problems under logical statement perturbation, indicating signs of memorization when solving these puzzles.

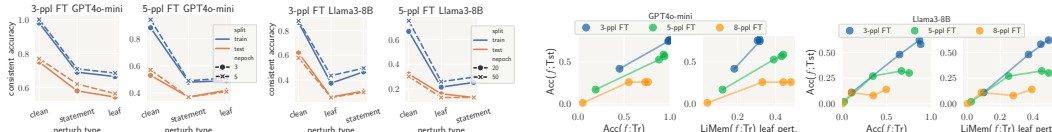

Figure 2: Accuracy of finetuned models drops under different perturbations. The drops on test set can be smaller than training set.

Figure 3: Test acc of FTed models increase with train acc, despite that the memorization becomes stronger with larger LiMem($f$; Tr) under leaf perturbation.

## 3.2 Fine-tuned Models

Here, we study the model's memorization capacity when directly fine-tuned on K&K problems. We take Llama3-8B and GPT4o-mini and run *supervised fine-tuning* (SFT) on a set of K&K training problems disjoint from the test set. Specifically, during SFT, the model observes the concatenation of the question and the answer for each problem, but the loss is only computed on the answer part.

**LLMs interpolate K&K training problems**. We fine-tune 50 epochs for Llama3-8B and GPT4o-mini for 5 epochs (due to budget constraints) via the OpenAI Finetune API (see details in § C). From Fig. 2 (clean), we observe high Acc($f$; Tr), and GPT4o-mini fine-tuned on 3-person puzzles reach interpolation (Acc($f$; Tr) $= 100\%$).

**Interpolating LLMs have large** LiMem($f$; Tr). In Fig. 2, we report the consistent accuracy Acc($f$; Tr) $\cdot$ CR under perturbation, defined as the ratio of samples correctly solved in both their original and perturbed forms. We observe significant gaps under math problem perturbations (e.g., statement and leaf) on training samples, suggesting that models have large LiMem($f$; Tr) and may rely on memorization to solve the training samples.

## 4 Large Language Interpolators Learn to Reason

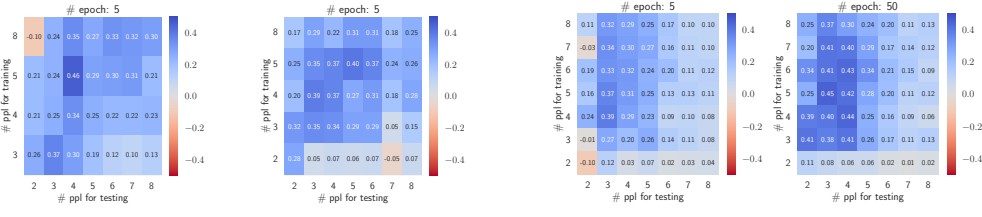

(a) GPT4o-mini CoT FT   (b) GPT4o-mini Direct FT          (c) Llama3-8B Direct FT

Figure 4: Test accuracy improvement on $N$-people problems for LLMs fine-tuned on $M$-people problems, compared to the unfine-tuned model, under 0-shot direct prompting. Most grid values are above 0, indicating transferability and enhanced reasoning abilities across unseen tasks. Results for more epochs are in **??**.

The studies in § 3 show that both off-the-shelf and fine-tuned models exhibit some level of memorization in solving K&K reasoning tasks. However, does it mean that those models do not have reasoning capabilities at all? It turns out that the models seem to do both, and the reasoning capability actually improves as the memorization level increases. Next, we present evidence that support this hypothesis.

**The generalization performance increases with memorization level**. As shown in Fig. 3, the accuracy of fine-tuned models on the test set continues to increase as Acc($f$; Tr) increases, despite that Lp$\Delta$ on training samples also increases (i.e., stronger memorization).

**The** LiMem($f$; Tst) **on test samples are smaller than** LiMem($f$; Tr) **on train samples** in Fig. 2, particularly for more challenging cases (e.g., 5-person puzzles). This suggests that models are more likely to use reasoning when solving unseen test samples.

**The fine-tuned model generalizes across different difficulty levels**. By fine-tuning on the $M$-person problem and testing on the $N$-person problem, we study LLMs' transferability. The $N \times M$ test accuracy improvement grid in Fig. 4 shows that 1) training on any $M$-person problem generally enhances accuracy on unseen $N$-person test problems for any $N$, indicating enhanced reasoning ability on both easier and harder problems; 2) extending the training epochs generally yields better results, particularly for Llama3-8B; 3) test accuracy improvement is larger when $N \leq 6$, and improving performance on more challenging tasks remains possible but more difficult.

## 5  Conclusion

In this paper, we designed a K&K puzzle-based logical reasoning benchmark and local perturbation-based metrics to quantify LLMs' memorization in reasoning tasks. Our results reveal an intriguing interplay between memorization and reasoning: while models heavily rely on memorization to solve challenging K&K puzzles, models trained to have a higher level memorization also solve more unseen test puzzles, and solve them relatively robustly (in contrast to the memorized training puzzles).

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

# Appendices

# A Related Work

**Memorization and benchmark contamination in LLMs**   Previous research has explored training data memorization in the context of privacy and copyright [6], focusing on how LMs may unintentionally reproduce text by generating near-verbatim outputs from their training data [16, 4, 5]. Our study broadens the concept of memorization to the reasoning context, by evaluating whether LLMs can recall solutions to training questions but struggle to solve their variants during testing.

Such memorization patterns appear in the off-the-shelf LLMs on popular math reasoning benchmarks, indicating potential benchmark contamination (i.e., included in the training data). For example, LLMs perform exceptionally well on benchmarks such as GSM8K, MATH, and MMLU, but their performance drops significantly when faced with benchmark variants. These include human-curated problems of similar difficulty [31], functional variants systematically generated via programs [23], rephrased versions [28], translated versions [29], or problems set beyond a specific date cutoff [22, 11].

**Logical reasoning benchmarks**   To evaluate logical reasoning capabilities in LLMs, synthetic benchmarks have been developed. These benchmarks enable scalable generation of samples with varying configurations and difficulty levels [9] to study LLM reasoning in a controlled setup. For instance, DyVal [32] dynamically generates evaluation samples with controllable complexity based on directed acyclic graphs, which covers reasoning tasks including deductive, boolean, and abductive reasoning. The authors demonstrate that fine-tuning Llama2-13B-chat on these synthetic samples enhances its performance on other reasoning benchmarks. Chen et al. [7] focus on propositional logic problems involving definite clauses. They synthetically generate variations with different premise orders, such as forward, backward, and shuffled. Their study shows that aligning the premise order with the proof order improves LLMs' accuracy in solving these problems. Dziri et al. [8] explore the limitations of transformers in tasks requiring compositional reasoning, including multiplication, Einstein's Puzzle (a constraint satisfaction problem), and dynamic programming problems. They find that while GPT-3 fine-tuned on their training samples can solve in-distribution problems, it fails to generalize to out-of-distribution tasks with increased problem sizes. Extending from Einstein's Puzzle, ZebraLogic [18] is introduced to require reasoning through reduction, absurdum, and elimination to solve constraint satisfaction problems. The study shows that off-the-shelf models struggle with complex puzzles involving large problem sizes. BoardgameQA[14] presents a question-answering dataset characterized by contradictory facts and rules in the questions. To solve this task, the authors find that fine-tuning BERT-large and T5-XXL on their training dataset with proofs outperforms few-shot prompting using PaLM with chain-of-thought (CoT) prompting. Alice in Wonderland [21] is a type of reasoning task in the format of light quiz-style problems such as "Alice has $N$ brothers and she also has $M$ sisters. How many sisters does Alice's brother have?" Advanced models often fail to reason and count accurately in such simple scenarios.

The closest benchmark to us is TruthQuest [19], which builds on the classic K&K puzzles but only focuses on evaluation samples involving 3-6 people puzzles. In contrast, our proposed framework extends this line of research by generating a more comprehensive set of K&K puzzles with more difficulty levels and various perturbation types. We evaluate state-of-the-art models on more challenging K&K puzzles (e.g., 8-people) and analyze their memorization behaviors (via fine-tuning) and generalization capabilities under local perturbations.

# B Data Generation Details

During our data construction, we use the maximum width $W = 2$ and depth $D = 2$, and the number of persons in the puzzle $N = 2, 3, 4, 5, 6, 7, 8$.

Tab. 1 presents the example knight (truth-teller) and knave (liar) scenario involving two people: Liam and Aria, with corresponding logical statements, and converted English statements, questions, and answers. It also shows three versions of the problems: an original example, a leaf-perturbed version, and a statement-perturbed version. Specifically, (1) leaf perturbation changes a "leaf" in the logical tree - a single truth value. In this case, it flipped Liam's status in Aria's statement from truth-teller to liar. (2) Statement perturbation changes the entire structure of a statement. Here, it changed Aria's statement from an equivalence to a contradictory conjunction. Both perturbations result in changing the answer. The leaf perturbation creates a subtle change in one statement that flips the logical outcome, while the statement perturbation changes the entire one statement.

Table 1: 2-person puzzle generation with the knight (telling-truth) and knave (lying) and comparison between original sample, leaf-perturbed sample, and statement-perturbed sample.

| Type | Example | Leaf Perturbed Example | Statement Perturbed Example |
|---|---|---|---|
| person | Liam (person index 0), Aria (person index 1) | | |
| logical statement | Liam: ('not', ('lying', 1)) Aria: ('⇔', ('telling-truth', 1), ('telling-truth', 0)) | Liam: ('not', ('lying', 1)) Aria: ('⇔', ('telling-truth', 1), ('lying', 0)) | Liam: ('not', ('lying', 1)) Aria: ('and', ('lying', 0), ('telling-truth', 0)) |
| English statement | "Aria is not a knave" - Liam. Aria asserted: "Aria is a knight if and only if Liam is a knight". | "Aria is not a knave" - Liam. Aria asserted: "Aria is a knight if and only if Liam is a knave". | "Aria is not a knave" - Liam. Aria asserted: "Liam is a knave and Liam is a knight" |
| question | "A very special island is inhabited only by knights and knaves. Knights always tell the truth, and knaves always lie. You meet 2 inhabitants: Liam, and Aria. Aria is not a knave - Liam. Aria asserted: Aria is a knight if and only if Liam is a knight. So who is a knight and who is a knave?" | "A very special island is inhabited only by knights and knaves. Knights always tell the truth, and knaves always lie. You meet 2 inhabitants: Liam, and Aria. Aria is not a knave - Liam. Aria asserted: Aria is a knight if and only if Liam is a knave. So who is a knight and who is a knave?" | "A very special island is inhabited only by knights and knaves. Knights always tell the truth, and knaves always lie. You meet 2 inhabitants: Liam, and Aria. Aria is not a knave - Liam. Aria asserted: Liam is a knave and Liam is a knight. So who is a knight and who is a knave?" |
| answer | (1) Liam is a knight (2) Aria is a knight | (1) Liam is a knave (2) Aria is a knave | (1) Liam is a knave (2) Aria is a knave |

## C  Experiments Details

**Evaluation**   We utilize zero-shot direct prompting with task-specific instructions for open-ended question-answering. We employ the following prompt:

---
**Prompt for 0-shot evaluation**

Your task is to solve a logical reasoning problem. You are given set of statements from which you must logically deduce the identity of a set of characters.

You must infer the identity of each character. At the end of your answer, you must clearly state the identity of each character by following the format:

CONCLUSION:
(1) ...
(2) ...
(3) ...

### Question: {question}
### Answer:

---

In our evaluation process, we use greedy decoding with temperature $t = 0$ for all models and a maximum token length of 2048.

To assess the correctness, we implement keyword matching: a response is considered correct if each person's ground truth identity appears in the conclusion part of the model's output.

**Fine-tuning**   For Llama3-8B fine-tuning, we employed the following standard hyperparameters with a batch size of 4, gradient accumulation steps of 8, and 5e-5 learning rate. We finetune for a maximum of 100 epochs.

For GPT4o-mini fine-tuning, we utilized the default hyperparameters provided by the OpenAI fine-tuning API. The model was fine-tuned for 5 epochs to achieve high accuracy within reasonable budget.

## D  Additional Experimental Results

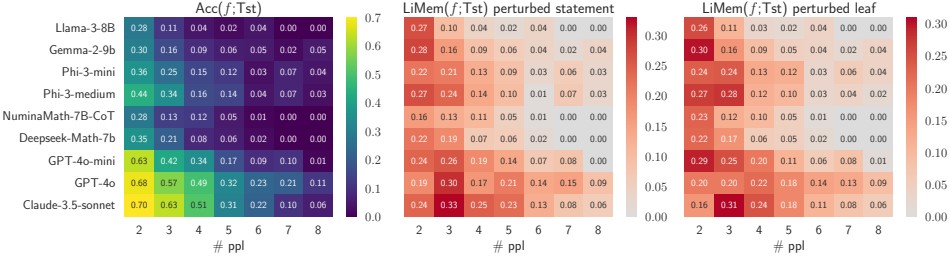

Figure 5: Test accuracy $\mathsf{Acc}(f;\mathsf{Tst})$ of off-the-shelf models under 0-shot direct prompting drops with increasing puzzle complexity (left). $\mathsf{LiMem}(f;\mathsf{Tst})$ on test examples under statement perturbation (middle) and leaf perturbation (right) is large for specific models, indicating signs of memorization in solving these puzzles.

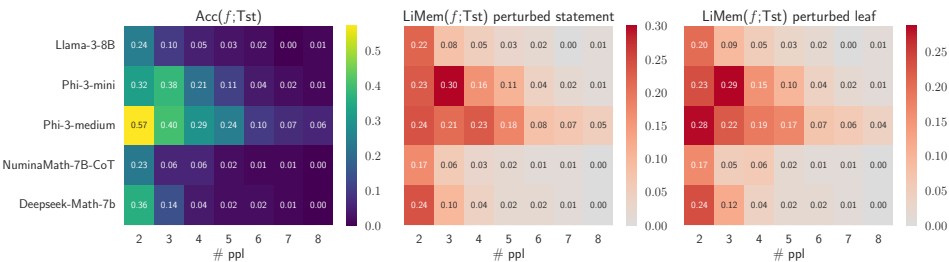

Figure 6: $\mathsf{Acc}(f;\mathsf{Tst})$ and $\mathsf{LiMem}(f;\mathsf{Tst})$ of off-the-shelf models under 0-shot CoT prompting, where we add a chain-of-thought trigger phrase " Let's think step by step" in the end of the prompt.

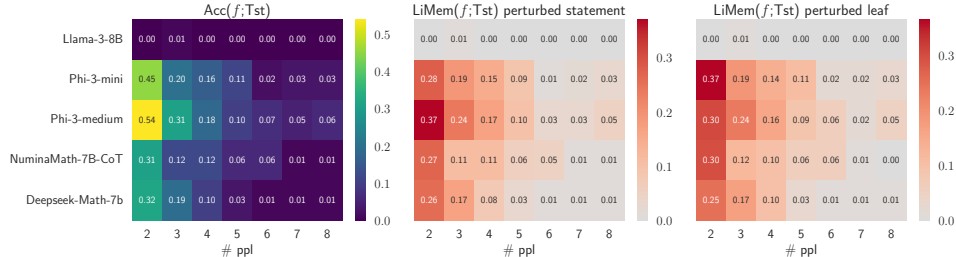

Figure 7: $\mathsf{Acc}(f;\mathsf{Tst})$ and $\mathsf{LiMem}(f;\mathsf{Tst})$ of off-the-shelf models under 1-shot direct prompting, where we provide one demonstration consisting of one question and its answer.

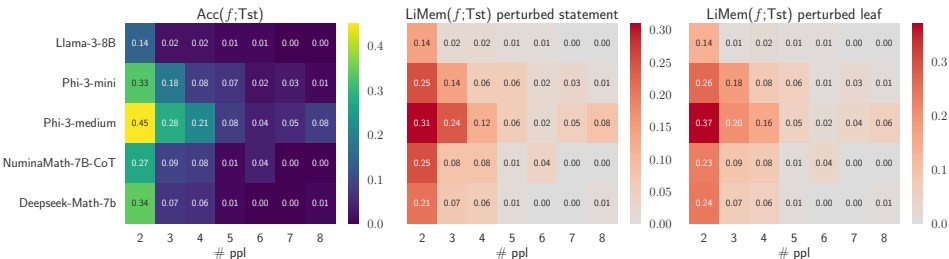

Figure 8: $\mathsf{Acc}(f;\mathsf{Tst})$ and $\mathsf{LiMem}(f;\mathsf{Tst})$ of off-the-shelf models under 1-shot CoT prompting, where we provide one demonstration consisting of a question, its corresponding CoT reasoning steps, and the answer.

