# OpenReview forum: "On Memorization of Large Language Models in Logical Reasoning"
_NeurIPS.cc/2024/Workshop/MATH-AI — MATH-AI 24_

### Official Review · Reviewer_L57y · 2024-09-28
**A Promising Approach to Understanding LLM Reasoning Through Knights and Knaves Puzzles**

**Rating:** 7
**Confidence:** 3

**Review:**

The paper explores how large language models (LLMs) balance memorization and reasoning in complex tasks. Using Knights and Knaves puzzles, the study shows that LLMs heavily rely on memorization to solve training puzzles but struggle when small changes are introduced. However, LLMs with higher memorization levels also perform better on unseen puzzles and are more robust under perturbation, suggesting an intricate relationship between memorization and genuine reasoning. This finding offers insight into LLM behavior and presents a path for future research.

# Pros:
## 1. Novel Benchmark: The paper introduces a Knights and Knaves (K&K) puzzle-based benchmark that dynamically generates logical reasoning problems, providing a fresh and controlled way to evaluate LLMs' reasoning abilities.

## 2. Memorization-Reasoning Interplay: It highlights an intriguing relationship between memorization and reasoning, showing that models with higher memorization levels can generalize better to unseen tasks and are more robust against perturbations.

## 3. Fine-Tuning Improvements: The study demonstrates that fine-tuning LLMs on these logical puzzles enhances both their reasoning and memorization capabilities, offering valuable insights into how training data impacts performance.

# Cons:
## 1. Lack of Comparative Analysis: The paper primarily focuses on its own benchmark without providing a strong comparison to other existing reasoning benchmarks. This makes it harder to evaluate how well the proposed method performs relative to other state-of-the-art techniques.

## 2. Limited Real-World Applicability: While the Knights and Knaves puzzles are a useful benchmark, they may not fully capture the complexity and diversity of real-world reasoning challenges, making it harder to generalize the findings to broader applications.

---

### Official Review · Reviewer_VKuL · 2024-10-05
**A good paper on LLM logical reasoning**

**Rating:** 7
**Confidence:** 4

**Review:**

This study examines the logical reasoning abilities of Large Language Models (LLMs) using Knights and Knaves puzzles as a case study. It introduces a bespoke benchmark for these puzzles and develops a method to quantify the impact of memorization in reasoning tasks. Findings reveal that LLMs, while achieving almost flawless results on training puzzles, struggle with minor modifications, indicating a significant reliance on memorization. However, the research also shows that higher levels of memorization enable LLMs to not only tackle more unfamiliar test puzzles effectively but also maintain their performance under modifications, suggesting a nuanced interplay between memorization and true reasoning skills.


Strength:

1. The paper innovates by developing a new benchmark based on Knights and Knaves puzzles, offering a novel way to assess LLMs' logical reasoning capabilities, especially under perturbed conditions.
2. The analysis delves deep into examining both the memorization capacity of LLMs and the relationship between memorization and reasoning. This dual focus provides new insights into the operational mechanisms of these models.


Weakness:
1. While challenging, the Knights and Knaves puzzles represent a very specific type of logic problem. This specialization might not sufficiently capture the full spectrum of logical reasoning abilities of LLMs.
2. The study focuses primarily on a specific type of puzzle and perturbation, which may not fully explore the models' capabilities across a broader range of logical problems

---

### Official Review · Reviewer_Ekeg · 2024-10-07
**Study on "Large Language Interpolators Can Learn Logical Reasoning: A Study on Knights and Knaves Puzzles"**

**Rating:** 7
**Confidence:** 4

**Review:**

Evaluation Overall

Divided Evaluation into Quality, Clarity and Originality

Quality: The paper presents a thoughtful examination of the interplay between memorization and reasoning in large language models (LLMs) using logical puzzles. The proposed LiMem metric and K&K benchmark offer a structured way to quantify this relationship. The experiments, particularly the fine-tuning on K&K puzzles and the introduction of perturbations, effectively showcase the balance between memorization and reasoning.

Clarity: The paper is well-written, with a clear and structured organization. However, further elaboration on the interpretation of LiMem scores would help readers less familiar with memorization in LLMs to better understand the implications.

Originality: The study is original, proposing a novel benchmark and a memorization metric for reasoning tasks. It builds upon existing work on memorization in LLMs, providing valuable insights into the role of memorization in generalization during reasoning.

Significance: The balance between memorization and reasoning is crucial for advancing LLMs. The K&K benchmark and the findings on memorization's role in reasoning contribute meaningfully to our understanding and improvement of LLM performance in practical applications.

-----

Pros
- Novel LiMem Metric: The new metric effectively measures memorization in reasoning, enhancing our understanding of LLM capabilities.
- K&K Benchmark: The logical reasoning benchmark with varying difficulty provides a robust and scalable evaluation framework.
- Insightful Interplay Analysis: The study demonstrates how memorization can enhance reasoning, offering an interesting perspective on LLM learning behavior.
- Thorough Model Evaluation: The tests on both pre-trained and fine-tuned models provide a comprehensive performance analysis.

-----

Cons
- LiMem Score Interpretation: The implications of LiMem scores could be further clarified to aid reader understanding.
- Generalization Beyond K&K: Broader reasoning benchmarks would enhance the study's generalizability beyond the K&K tasks.
- Model Complexity and Scaling: Deeper exploration of the effect of model size on memorization and reasoning would improve the overall analysis.

---

### Official Review · Reviewer_1iFW · 2024-10-08
**Review of the paper**

**Rating:** 7
**Confidence:** 3

**Review:**

Pros:
This paper explores whether LLMs depend on memorization or genuine reasoning when tackling logical puzzles, using a newly introduced benchmark based on Knights and Knaves puzzles. The authors introduce the Local Inconsistency-based Memorization Score (LiMem), a metric that measures the extent of memorization by evaluating performance on both training puzzles and altered (unseen) versions. Interestingly, the findings reveal that while the models significantly rely on memorization to solve training puzzles, they also develop enhanced reasoning abilities, improving their capability to solve new puzzles and adapt to variations with greater resilience. The results suggest an intricate interplay between memorization and reasoning in LLMs, indicating a relationship that warrants further exploration to fully comprehend the models' reasoning mechanisms.

Cons:
Relying on a single type of dataset may limits the generalizibility of the findings. Incorporating a variety of logical reasoning benchmarks would provide a more comprehensive understanding across various contexts.

---

### Decision · Program_Chairs · 2024-10-07

Accept